# Are Gait Patterns during In-Lab Running Representative of Gait Patterns during Real-World Training? An Experimental Study

**DOI:** 10.3390/s24092892

**Published:** 2024-05-01

**Authors:** John J. Davis, Stacey A. Meardon, Andrew W. Brown, John S. Raglin, Jaroslaw Harezlak, Allison H. Gruber

**Affiliations:** 1Department of Kinesiology, School of Public Health-Bloomington, Indiana University, Bloomington, IN 47405, USA; raglinj@indiana.edu; 2Department of Physical Therapy, East Carolina University, Greenville, NC 27858, USA; meardons@ecu.edu; 3Department of Biostatistics, University of Arkansas for Medical Sciences, Little Rock, AR 72205, USA; awbrown@uams.edu; 4Department of Epidemiology and Biostatistics, School of Public Health-Bloomington, Indiana University, Bloomington, IN 47405, USA; harezlak@iu.edu

**Keywords:** wearable technology, depth statistics, unsupervised learning, free-living gait, biomechanics

## Abstract

Biomechanical assessments of running typically take place inside motion capture laboratories. However, it is unclear whether data from these in-lab gait assessments are representative of gait during real-world running. This study sought to test how well real-world gait patterns are represented by in-lab gait data in two cohorts of runners equipped with consumer-grade wearable sensors measuring speed, step length, vertical oscillation, stance time, and leg stiffness. Cohort 1 (*N* = 49) completed an in-lab treadmill run plus five real-world runs of self-selected distances on self-selected courses. Cohort 2 (*N* = 19) completed a 2.4 km outdoor run on a known course plus five real-world runs of self-selected distances on self-selected courses. The degree to which in-lab gait reflected real-world gait was quantified using univariate overlap and multivariate depth overlap statistics, both for all real-world running and for real-world running on flat, straight segments only. When comparing in-lab and real-world data from the same subject, univariate overlap ranged from 65.7% (leg stiffness) to 95.2% (speed). When considering all gait metrics together, only 32.5% of real-world data were well-represented by in-lab data from the same subject. Pooling in-lab gait data across multiple subjects led to greater distributional overlap between in-lab and real-world data (depth overlap 89.3–90.3%) due to the broader variability in gait seen across (as opposed to within) subjects. Stratifying real-world running to only include flat, straight segments did not meaningfully increase the overlap between in-lab and real-world running (changes of <1%). Individual gait patterns during real-world running, as characterized by consumer-grade wearable sensors, are not well-represented by the same runner’s in-lab data. Researchers and clinicians should consider “borrowing” information from a pool of many runners to predict individual gait behavior when using biomechanical data to make clinical or sports performance decisions.

## 1. Introduction

Individual differences in running biomechanics have been associated with both injury and performance [1,2,3,4]. Running gait has traditionally been assessed during in-lab motion capture sessions, but data collected under these conditions are only useful for real-world clinical and sporting applications if the in-lab data are well-representative of gait patterns adopted during real-world training. Since virtually all training takes place outside of the lab, the generalizability of models or inferences based on in-lab biomechanical data to real-world running is of paramount importance for both clinicians and researchers. Likewise, generalizability of in-lab data presents challenges in professional sport, for similar reasons: gait characteristics in matches or competitions may not reflect those seen during in-lab evaluations conducted as part of training or injury rehabilitation.

While a recent systematic review found only minor biomechanical changes when comparing gait patterns during treadmill versus overground running [5], other work that more directly compares in-lab to real-world running has noted changes in various aspects of running gait. Lafferty et al. report that video-based gait analysis showed differences in gait variables including footstrike angle, tibial inclination, and pelvic drop when comparing indoor treadmill versus outdoor track running [6], and Benson et al. developed a classifier based on sensor-measured gait features that could differentiate between treadmill and sidewalk running with ~80% accuracy, suggesting distinctive differences in the characteristics of gait in the lab versus in the real world [7]. To date, though, the previous literature has focused on differences in the mean value of individual gait metrics, and has not considered how overall gait patterns are distributed during in-lab and real-world running. Moreover, whether differences seen in real-world running can be ascribed to changes in the environment (e.g., turns, inclines, declines) remains unclear. 

Consumer-grade sensors are particularly attractive for real-world gait assessment because of their low cost, wide usage, and ability to synchronize data with cloud-based training platforms, which allows researchers and clinicians to remotely collect and monitor gait data on hundreds or thousands of runners at once [8]. Recent work has explored using both research-grade and consumer-grade wearable sensors to characterize gait patterns during real-world running, due to the ease with which wearable sensors can be used outside of the lab [9,10]. 

Measuring a runner’s full gait pattern during real-world running is challenging despite the utility of wearable sensors. Both consumer-grade and research-grade wearable sensors measure only a limited number of gait metrics compared with what is possible with in-lab motion capture equipment, and not all devices measure the same gait metrics. However, using multiple devices together can capture gait metrics such as speed, stride length, vertical oscillation, ground contact time, and leg stiffness. Many of these same gait metrics are used in simplified biomechanical models of running, such as the well-studied mass-spring model, which explains many key aspects of running biomechanics [11]. Combining these gait metrics gives rise to the idea of a “gait pattern”—a set of gait metrics that jointly represent the body’s movement. Comparing sensor-measured gait patterns between runners or between different conditions (e.g., in-lab versus real-world) is a straightforward way to quantify similarities or differences in gait.

To this end, the primary goal of this study was to compare the distribution of gait patterns during in-lab and real-world running. Three related questions are relevant when considering whether gait patterns during in-lab running are representative of gait patterns during real-world running. 

First, is a runner’s gait pattern during an in-lab gait analysis a good representation of that same runner’s real-world gait pattern? This question is relevant to in-lab biomechanical analyses and in-lab gait retraining interventions, which are done with the aim of generalizing from in-lab running to real-world running in the same individual. 

Second, is a set of in-lab gait data from a large pool of runners a good representation of the real-world gait pattern that might be observed in a new runner from the same population? This question is relevant when constructing predictive models based on in-lab data that aim to generalize to real-world running data from new, unseen runners (i.e., runners whose data were not used to develop the model). For example, Matijevich et al. [12] developed a sensor-based model for predicting compressive forces on the tibia. Successfully applying this model to free-living data from runners in the same population would require the in-lab data collected on the subjects who formed the “training set” to be a good match for the real-world running data from a new, unseen “test” subject. 

Third, is a set of in-lab gait data from a large pool of runners a good representation of the real-world gait pattern that might be observed from a new runner from a new population, potentially in a different geographic location? This question is relevant when discussing the translatability of study findings, i.e., whether statistical inferences or predictive model performance from a study on one population of runners (e.g., healthy adults in one location) will generalize to a different or more specialized population of runners (e.g., college-aged females in a different location). For example, this question would be important for clinicians who want to apply findings from a published study to the real-world training of a patient from a new population, and for researchers who want to apply a published predictive model to a new sample of runners.

This study addressed each of these questions by quantifying the degree of overlap between gait patterns during in-lab and real-world running, as measured by a set of consumer-grade wearable sensors. Further, this study disaggregated the effects of the real-world running environment (inclines, declines, and turns) from changes in gait pattern on flat, straight settings by comparing gait patterns during all real-world running, versus real-world running only on flat, straight segments. 

## 2. Materials and Methods

### 2.1. Overview

This study involved two separate cohorts of runners representing different populations of potential interest to researchers and clinicians. Cohort 1 consisted of healthy male and female runners aged 18 and older who completed an in-lab treadmill run while equipped with a set of consumer-grade wearable sensors. These same runners completed five real-world, free-living runs using the same set of sensors. Cohort 2 followed a different protocol, which was designed to assess the generalizability of the findings from Cohort 1 to a new population, as well as to determine the potential sources of gait differences between in-lab and real-world running, specifically, the influence of turns, inclines, and declines. Cohort 2 consisted of healthy female runners aged 18 and older in a different geographic location who completed a 2.4 km run on a measured course with known segments of flat, turning, incline, and decline running while wearing the same set of sensors as Cohort 1. This cohort also completed five real-world, free-living runs, again using the same set of sensors.

Gait metrics from the wearable sensors were used to characterize the gait pattern for each runner, and the distributions of these metrics during in-lab and real-world running were compared to quantify the proportion of overlap in gait patterns across these distributions. 

### 2.2. Participants

**Cohort 1.** The inclusion criteria for Cohort 1 were designed to capture a pool of runners representative of the broader population of runners. Healthy runners aged 18 and older were recruited, with no upper limit on age. Participants were required to run at least three times per week with at least one run of 40 min or longer, were required to have no current musculoskeletal injury that prevented them from doing their usual running training, and were required to meet American College of Sports Medicine preparticipation guidelines for exercise [13]. Runners were recruited from the community via social media, flyers at local running stores, and in person recruitment at local running events. Recruitment and data collection for Cohort 1 took place in Greenville, North Carolina, which is located in a region with predominantly flat terrain. All participants provided written informed consent, and the study was approved by the East Carolina University and Medical Center IRB and the Indiana University IRB (protocols # 21-001137 and 12040). The sample size for Cohort 1 was determined via a learning curve power analysis for a predictive modeling goal detailed elsewhere [14], which indicated that a minimum of 40 participants were needed.

**Cohort 2.** The inclusion criteria for Cohort 2 were designed to construct a more homogenous and specialized population of runners to assess the generalizability of in-lab data to a new population of athletes. One such specialized population often studied in prospective research on running injuries is young adult female runners, who may be at greater risk of overuse injury (e.g., Davis et al., Rauh et al. [15,16]). In service of this goal of testing the generalizability of findings from the in-lab data, Cohort 2 included women aged 18–32 were recruited who fulfilled the same inclusion criteria as Cohort 1 (running at least three times per week with one run lasting at least 40 min, and no current injuries or contraindications for exercise). Runners were recruited from students at a large university via flyers on campus and at a local running club. Recruitment and data collection for Cohort 2 took place in Bloomington, Indiana, which is located in a region with predominantly hilly terrain. All participants provided written informed consent, and the study was approved by the Indiana University IRB (protocol #17923). The sample size for Cohort 2 was designed to recruit a similar number of female subjects as were recruited for Cohort 1. 

### 2.3. Wearable Sensors and Gait Metrics

Three consumer-grade wearable sensors were used to collect gait data during in-lab and real-world running: a sports watch with global navigation satellite system (GNSS) capabilities (Garmin Forerunner 245, Garmin Ltd., Olathe, KS, USA) worn on the participant’s left wrist; a chest strap heart rate monitor with an integrated accelerometer (Garmin HRM-Run and HRM-Tri, Garmin Ltd., Olathe, KS, USA), which was worn around the chest, centered over the heart and inferior to the sternum, and a foot pod with an integrated inertial measurement unit (Stryd v2, Stryd Inc., Boulder, CO, USA), which was placed on the distal shoelaces of the left shoe. This combination of devices was chosen because these devices are already in wide use, record and synchronize their data to remote cloud-based training platforms, and capture key biomechanical aspects of gait that can be used to characterize a runner’s gait pattern. Six separate matched sets of these three devices were used to reduce any device-specific systematic errors, and to enable parallel enrollment of multiple subjects. Three of these device sets used the HRM-Run model of chest strap sensor, and three of these devices sets used the HRM-Tri model of chest strap sensor; using multiple variants of this device (both of which are in wide use) expanded the real-world generalizability of predictive models built as a separate part of the project [14]. Validation testing on a separate cohort of ten runners showed that the chest strap-measured gait metrics show close agreement in the gait metrics measured across the two variants of the chest strap (see Appendix A, which details mean absolute percentage differences between devices).

Five sensor-measured gait metrics were selected to represent a runner’s gait pattern: running speed, step length, vertical oscillation, stance time, and leg stiffness. These five specific gait metrics were selected because they correspond to key parameters of the mass-spring model of running, a simple and well-studied model that describes numerous aspects of running gait [11], and because these specific metrics are measured with acceptable accuracy by the wearable sensors. Step length and vertical oscillation were measured with the chest strap, while stance time and leg stiffness were measured by the foot pod.

Since the GNSS technology of the sports watch does not work indoors, speed data from the foot pod was used to measure running speed on the treadmill. Both the foot pod and GNSS speed estimates have errors of <2% compared to ground-truth running speed in previous research, and the foot pod’s speed data showed no statistically significant bias against the ground-truth treadmill speed (see Appendix A) [17,18]. 

The accuracy of the individual gait metrics was determined empirically for runners in Cohort 1 using motion capture data collected during the in-lab treadmill run (See Appendix A for full device metric validation results including correlation coefficients and Bland–Altman limits of agreement). Though not all metrics are measured by the devices with equal absolute accuracy, this study’s validation and other validation studies on the same devices have determined that these metrics are measured with sufficiently high accuracy compared to in-lab metrics to detect changes in gait within and across individuals [19,20,21].

In the case of gait metrics measured by two sensors, the sensor which measured that gait metric more accurately was used—this was the chest strap for vertical oscillation, and the foot pod for stance time. Since speed, cadence, and stride length are mathematically linked, only speed and stride length were used in the representation of a runner’s gait pattern, because (1) runners often vary their speed by changes in stride length as opposed to cadence [22,23], and (2) the devices quantize cadence by mapping it to an integer number of strides (foot pod) or an integer number of steps (chest strap) per minute. This quantization process introduces errors compared to using stride length, which is measured by the chest strap to the millimeter. 

During all runs, the chest strap and foot pod streamed their data wirelessly to the sports watch, which recorded the gait metric values once per second alongside GNSS-determined latitude, longitude, and speed (during outdoor running). All gait metrics for each running session were saved in a single Flexible and Interoperable Data Transfer (FIT) protocol file.

### 2.4. Protocol

**Cohort 1, in-lab run.** Participants in Cohort 1 first completed a 38 min in-lab treadmill run at speeds ranging from 30% slower to 25% faster than each runner’s self-reported preferred running speed for a “typical training run.” This range of speeds was designed to increase the variability in each runner’s gait as observed in the lab, as most gait-related parameters change as a function of speed. The range of speeds was selected by comparing data on self-reported preferred running speed from a previous in-lab study [24] with known values for the typical walk–run transition speed in healthy adults [25] and predictive equations for estimating lactate threshold from training pace [26]. The range of 30% slower to 25% faster kept the slowest speeds above the walk–run transition for most adults, avoiding uncomfortably slow speeds, and kept the fastest speeds below each runner’s predicted lactate threshold, avoiding excessive fatigue. The speeds were presented in a semirandomized fashion, with the slowest two speeds first, then a block of randomized speeds, followed by the fastest two speeds at the end. This ordering was chosen to maximize the range of speeds covered by each subject and to minimize early-onset fatigue that would prevent subjects from completing the protocol. To minimize any potential order effects, each subject was randomly assigned one of four randomized block orders (speed ordering for each protocol provided in Appendix A). The in-lab treadmill run, which took place as a part of a larger study [14], was completed in a motion capture lab while equipped with the wearable sensors. 

**Cohort 2, measured course run**. Participants in Cohort 2 first completed a 2.4 km run on a measured out-and-back course while under observation by research staff and while equipped with the wearable sensors. The measured course consisted of known segments of flat and straight running, left turns, right turns, inclines, and declines (Figure 1), all of which were confirmed via mapping software (OpenStreetMap, accessed 28 April 2023). The flat and straight segment was a portion of a concrete running track with inclines and decline magnitude of <0.5% grade. The left and right turns were turning portions on this same track. The incline and decline segments were on a paved sidewalk with an average grade of 5.5%.

**Cohorts 1 and 2, real-world runs.** After completing the in-lab or measured course run, participants in Cohort 1 and Cohort 2, respectively, were sent home with the wearable sensors. Both cohorts were instructed to record five outdoor runs, with no restrictions on the course, terrain, run distance, or pace. In this way, the data from the real-world runs were designed to be a representative sample of the participants’ typical gait patterns during their typical day-to-day training. For both cohorts, five real-world runs were selected because previous research has shown that this number of runs is sufficient to generate a stable characterization of the distribution of a runner’s gait pattern during real-world running [27].

### 2.5. Data Extraction

Wearable sensor data from all conditions and cohorts were downloaded as .FIT format files. For the measured course run and all real-world runs, GNSS-based location data were downloaded as GPS Exchange Format (GPX) files from the Garmin Connect online training platform. Elevation data in the GPX files were derived from GNSS location queries to a database of professional land survey data. FIT and GPX data were aligned via timestamps, which were both derived from the on-device clock on the sports watch.

**Cohort 1, in-lab run.** Sensor data from each of the 12 running trials were trimmed to exclude the first 25 s and the last 15 s of each segment to remove any effects caused by changes in the treadmill belt speed. During these portions of the protocol, the gait metrics recorded by the devices lag behind the runner’s true gait metrics because the treadmill belt speed is not constant—belt speed changes gradually over several seconds to avoid causing a trip hazard.

**Cohort 2, measured course run**. GNSS-based position data were cross-referenced against latitude-longitude bounding boxes to extract segments of the measured course run that took place within the known segments of flat and straight running, left turns, right turns, inclines, and declines (Figure 1). These segments were identified using a ground-truth course measured using mapping software and elevation data from OpenStreetMap.

**Cohorts 1 and 2, real-world runs.** Since real-world running data contain some amount of standing and walking, portions of the real-world data which had speeds below 1.56 m/s or a cadence below 100 strides per minute were excluded, following similar strategies used in previous work [27,28]. These strategies resulted in 6.5% of real-world data being excluded. 

### 2.6. Data Processing

To identify straight versus turning portions of real-world running, turn rate was calculated using a central-difference derivative applied to GNSS-based position data. Details of this central-difference scheme are described in Appendix A. An example of turn rate data is shown in Figure 2 for a portion of the known course run.

### 2.7. Identification of Flat and Straight Segments in Real-World Running

After calculating the turn rate and incline/decline of all real-world data, flat and straight segments were identified by using cut-off thresholds for both turn rate and incline/decline. These thresholds were determined using data from the flat, straight segment of the known course run completed by runners in Cohort 2. Using the calculated turn rate and incline/decline data from this segment, thresholds were identified which retained >99% of datapoints from this known flat, straight segment. Applying these criteria led to a turn rate threshold of 6.34 degrees per second and an incline/decline threshold of 2.28% grade (Figure 3). Notably, this empirically-derived grade threshold is very similar to the 2% grade threshold used to identify flat portions of running in Benson et al. [27]. These thresholds are nonzero because of the inherent jitter in GNSS data caused by atmospheric conditions, buildings, and other sources of interference. As a consequence of this jitter, even running on a perfectly flat and straight segment can result in nonzero calculated incline/decline and turn rate data.

### 2.8. Statistical Comparison of Gait Patterns

Two methods were used to quantify distributional overlap between sets of gait patterns (e.g., in-lab and real-world gait patterns): a univariate analysis and a multivariate depth analysis. 

**Univariate analysis.** The univariate analysis treated each gait metric separately. For each gait metric and each comparison, the overlap between a reference distribution D and a new distribution $D^{*}$ was calculated as follows:Calculate the central 95% range of the points in D. This corresponds to the [2.5%, 97.5%] quantiles of this gait metric in this distribution.Calculate the proportion of points in $D^{*}$ which fall within this central 95% range calculated from D. Data points in $D^{*}$ which fall within this central 95% range were considered to be well-represented by the reference distribution D (Figure 4). 

**Depth analysis.** The univariate analysis cannot detect changes in how multiple gait metrics covary together. For example, if the speed–stride length relationship differs for a particular subject inside the lab versus outside the lab, this shift would not be fully captured by the univariate analysis. The multivariate approach using depth statistics was used to address this shortcoming.

The concept of statistical depth quantifies how “deep” in a multivariate distribution (or point cloud) a given point is, relative to the rest of the data in the K-dimensional point cloud, where the number of dimensions (K) is the number of variables of interest [29]. The depth of a data point in a point cloud is a measure of how centrally-located it is with respect to the rest of the distribution, and can be thought of as a multivariate generalization of the median. This study used Tukey’s half-space depth because of its simplicity, interpretability, and the availability of fast approximation methods that are scalable to large datasets (such as the ~500,000 data points collected during Cohort 1′s real-world training) [29]. The half-space depth of a point d with respect to a point cloud D is calculated as the smallest possible fraction of all data points in a point cloud that can be separated from the remaining points by a half-space that contains point d (illustrated in Figure 5). A half-space is the portion of K-dimensional space that lies on one side of an K−1 dimensional hyperplane in K-dimensional space. Concretely, Figure 5 shows points in two dimensions; a one-dimensional line partitions two-dimensional space into two half-spaces.

This depth analysis method was used to assign a depth to every point in a point cloud D (e.g., the in-lab gait data from one subject), with the depth of each point quantifying how well-represented that point is with respect to the overall data distribution. The depth of a new point $d^{*}$ from a new point cloud $D^{*}$ (for example, the real-world gait data from the same subject) was calculated relative to the original point cloud D. In this way, the depth of a data point $d^{*}$ with respect to D quantifies how well-represented that new observation is compared to the reference data distribution.

Within the original data distribution D, there exists a convex hull that contains 95% of all of the data points in D, termed the 95% depth region, just like the [2.5%, 97.5%] quantiles from the univariate analysis contain 95% of all observations of a single variable. The depth value that denotes the minimum depth of this 95% depth region is termed the 95% depth cut-off, and is calculated by finding the cumulative 95% quantile of the depth of all data points in D. As with the univariate analysis, this study considered this 95% depth cutoff as denoting the space of gait patterns that can be considered “well-represented” in the reference distribution.

The depth analysis proceeded as follows, using the example of the comparison of in-lab gait patterns from one runner to the real-world gait patterns from that same runner. In this example, the gait pattern is a five-dimensional point d which contains the five sensor-measured gait metrics.Define the reference distribution of gait patterns, D (e.g., a set of five-dimensional vectors containing all in-lab gait data from one subject in Cohort 1).Calculate the half-space depth for all points d ∈D, with respect to the reference distribution D.Calculate the 95% depth cutoff by finding the 95% quantile value of the depth values for all points in D.Define the new distribution of gait patterns, $D^{*}$ (e.g., real-world data from the same subject in Cohort 1).Calculate the half-space depth for all points $d^{*}\in D^{*}$, with respect to the distribution D.Calculate the proportion of points in $D^{*}$ that are as deep or deeper than the 95% depth cutoff determined in step 3.


The proportion of points that are as deep or deeper than the 95% depth cutoff is the proportion of points in $D^{*}$ which are well-represented by the distribution of points in D. In the example from above, this would be the proportion of data collected during real-world running that is well-represented by the in-lab gait data for that subject. This concept is illustrated in a simple two-dimensional example in Figure 5. Because this analysis uses five gait metrics to represent a runner’s gait pattern, the actual depth analysis takes place in five-dimensional space (K=5). The depth analysis provides a single number summarizing the joint distribution of the runner’s gait patterns (as characterized by the set of five-dimensional vectors containing each of the five gait metrics chosen) for each comparison.

**Gait distributions comparisons and summary statistics.** To quantify similarities between in-lab and real-world gait patterns, the gait pattern distribution comparisons listed in Table 1 were made using both the univariate and depth analysis approaches outlined above. The comparisons listed in Table 1 address the questions posed in the introduction, which examine the degree to which in-lab gait is representative of real-world gait for the same runner, a runner from the same population of participants, and a runner from a different population.

Each comparison was carried out in a subject-wise fashion, operating similarly to leave-one-subject-out cross-validation. For each comparison, and each quantification method (univariate and depth-based multivariate), means across subjects and 95% confidence intervals to assess uncertainty were calculated using nonparametric bootstrapping with 10,000 replicates, as implemented in the R package ‘boot’ version 1.3-28 [30]. This bootstrapping approach addresses differences in the sample size between Cohort 1 and Cohort 2 by accurately reflecting the increased uncertainty in comparisons made using Cohort 2, which had fewer participants. Depth analysis was calculated using the ‘depth.halfspace’ function from the R package ‘ddalpha’ version 1.3.13 [31] using the random approximation method of Cuesta-Albertos and Nieto-Reyes [32] with 1000 random directions. To make depth calculation computationally tractable, real-world data were thinned by taking a random subsample of 25% of the data [32].

## 3. Results

### 3.1. Participants, Recruitment, and Running Data Summary

In Cohort 1, 60 runners were recruited for participation. Of these, eleven subjects were excluded for the following reasons: two were uncomfortable running on the treadmill, one did not complete the real-world portion of the study, four experienced device malfunctions during either the in-lab or real-world portion of the study, one sustained an ankle sprain (unrelated to study procedures) after the in-lab run but prior to completing the real-world runs, and three completed no runs with a GNSS connection, leaving 49 participants in Cohort 1 who recorded a total of 247 running sessions (some participants recorded individual training session as multiple runs on the watch, e.g., a warm-up, a workout session, and a cool-down). Twenty-four of these running sessions were excluded because they did not include GNSS data (querying the participants revealed that the cause was beginning a run prior to GNSS satellite acquisition on the watch; improving the instructions given to participants reduced the rate of no-GNSS-data runs in subsequent data collections), and three of these running sessions were excluded because the foot pod or chest strap was not worn for the run. 

After these exclusions, 219 runs (149.08 h of running) were included in the analysis. Participant demographics are provided in Table 2. In Cohort 1, 77.32% of running time took place on flat segments, 86.52% took place on straight segments, and 67.22% of running took place on segments that were flat and straight. 

In Cohort 2, 23 runners were recruited for participation. Of these, 4 were excluded because they did not have time to complete real-world runs prior to the study’s conclusion, yielding 19 runners who recorded 95 running sessions. Six of these running sessions were excluded because they did not include GNSS data. After these exclusions, 89 runs (47.97 h) were included in the analysis. In Cohort 2, 55.57% of running time took place on flat segments, 87.35% took place on straight segments, and 48.58% took place on segments that were flat and straight. The difference in proportion of flat segments was expected based on geographic differences between the data collection locations in Cohort 1 and Cohort 2—the terrain in the geographic location of Cohort 2 was much hillier than for Cohort 1. The small number of self-reported elite runners in both cohorts were not noticeable outliers in any of the analyses.

### 3.2. Univariate Analysis of Gait Pattern Overlap

**Analysis 1.** When using in-lab data from one subject as the reference distribution, real-world data from that same subject showed overlap from 65.7 to 95.2% on average across gait metrics, but with some subjects displaying overlap below 50% (Figure 6A). Average overlap was lowest for vertical oscillation (74.5%) and leg stiffness (65.7%), indicating a change in these gait metrics when a subject was running in-lab versus in the real-world. 

**Analysis 2.** When using “leave-one-subject-out” in-lab data from Cohort 1 as the reference distribution, real-world data were well-represented by the in-lab data. Overlap for speed (98.3%), step length (96.6%), vertical oscillation (92.4%), stance time (97.3%), and leg stiffness (91.3%) were all close to 95%, indicating strong overlap between distributions. However, one to five outlying subjects had distributional overlap below 50%, with some near zero (Figure 6B)—these individuals had marked differences in their real-world step length, ground contact time, and leg stiffness, which drove this poor overlap.

**Analysis 3.** When using in-lab data from Cohort 1 as the reference distribution, real-world data from Cohort 2 were well-represented by the in-lab data for speed (95.9%), step length (94.7%), vertical oscillation (99.4%), and stance time (97.9%), but to a lesser extent leg stiffness (88.3%), which had lower mean values because of three subjects with lower distributional overlap (Figure 6C).

**Analysis 4.** When using real-world data from Cohort 1 as the reference distribution, real-world data from Cohort 2 were well-represented by Cohort 1′s real-world data for speed (93.0%), step length (94.0%), vertical oscillation (99.6%), stance time (88.6%), and leg stiffness (95.0%), albeit with lower overlap for two subjects (Figure 6D).

**Stratification by flat and straight segments.** For all analyses, overlap changed by less than five percentage points when stratifying real-world data to include only flat, straight segments (red vs. blue points in Figure 6).

### 3.3. Depth Analysis of Gait Pattern Overlap

**Analysis 1.** When using in-lab data from one subject as the reference distribution, real-world data from the same subject showed an average of 32.5% overlap (Figure 6A). For ten subjects, overlap was less than 10%. 

**Analysis 2.** When using in-lab data from Cohort 1 as the reference distribution, real-world data from a new subject from Cohort 1 showed overlap of 89.5%, though the confidence intervals for this average overlap excluded 95%—the overlap that would be expected for data drawn from the same underlying distribution, because overlap criteria were set as falling within the central 95% of the reference distribution (Figure 6B). 

**Analysis 3.** When using in-lab data from Cohort 1 as the reference distribution, real-world data from Cohort 2 overlapped 90.3% with the in-lab data, with confidence intervals including 95%. One outlying individual had overlap near zero, caused by running slower and with shorter steps than most of the data in Cohort 1 (Figure 6C).

**Analysis 4.** When using real-world data from Cohort 1 as the reference distribution, real-world data from Cohort 2 showed overlap of 91.6%, with confidence intervals including 95%. One outlying individual with zero overlap; this individual ran slower and with shorter steps than any of the runners in Cohort 1 (Figure 6D).

**Stratification by flat and straight segments.** As with the univariate analysis, stratification of real-world data to include only flat, straight segments resulted in less than a five-percentage point change for all the analyses (red vs. blue points in Figure 6), indicating that the turns, inclines, and declines encountered during real-world running were not the primary driver of differences between in-lab and real-world gait patterns.

### 3.4. Sensitivity Analysis of Gait Pattern Metric Choice

A sensitivity analysis demonstrated that the decreased overlap in the depth-based analysis could not be attributed to any one gait metric, nor could it be explained by changes in the speed–step length relationship (Figure 7). Even when gait patterns were characterized using only speed, cadence, and ground contact time, in-lab versus real-world overlap remained below 50%.

## 4. Discussion

The main objective of this study was to assess to what degree gait patterns during real-world running are well-represented by in-lab running gait. Individually, speed, step length, and stance time are largely similar when moving from the lab to the real-world, based on results from the univariate analysis. However, less than one-third of steps taken during real-world running are well-represented by a runner’s own in-lab data when these gait metrics are considered together using depth analysis. This finding indicates that the relationships between gait metrics is changing, rather than their univariate range. This fraction is materially unchanged (<5%) when restricting real-world running to flat, straight segments. 

### 4.1. Distributional Shifts in Real-World Data

Together, these findings indicate that real-world gait often exhibits a distributional shift (as illustrated in Figure 8), and as a consequence, many of the steps taken by a runner during real-world training use gait patterns that are never seen during an in-lab data collection, even when in-lab data are collected at similar speeds as the real-world training. As such, efforts to use in-lab data to create athlete-specific, sensor-based models of gait may face difficulties when attempting to apply these models to real-world data, since the same athlete runs differently in the lab versus in the real world.

The overlap of ~90% seen in the depth analysis for Analyses 2 and 3 indicates that pooling in-lab data from a large number of runners can better-represent real-world gait patterns, even when the new runner comes from a different population than the in-lab reference data. Since these comparisons were made using subject-wise cross-validation, a subject’s own in-lab data were not included in the pool of in-lab data used to make overlap comparisons. Thus, the improvements compared to Analysis 1 indicate that aggregated in-lab data from many runners can act as a stand-in for real-world data from a new runner. A dataset with many runners contains much more variability in gait pattern than a dataset from a single runner, thus increasing the probability that any given step taken by a runner in the real-world has a “match” within the pool of in-lab gait patterns. 

Collectively, the findings from Analyses 1, 2, and 3 suggest that experimental results that rely on in-lab data will be more likely to generalize to real-world running if they are drawn from a large dataset of in-lab data, pooled across many runners, even if the target application is a subject-specific model or prediction. The increase in overlap from Analysis 1 to Analysis 2 is a strong example of this principle at work.

### 4.2. Effects of Inclines, Declines, and Turns

The overlap between in-lab and real-world gait patterns was trivially affected by stratifying real-world running to only include running on flat, straight segments. Overlap was often greater, but only by 0.8% or less. This finding indicates that the differences between in-lab and real-world gait patterns seen in this study cannot be solely attributed to inclines, declines, and turns: runners run differently in the real world versus in the lab, even when running on flat, straight segments. Notably, other work has shown that sensor-measured tibial acceleration differs between in-lab treadmill running and real-world running on flat, straight ground [34], as well as between in-lab overground running and real-world running on flat, straight ground [35].

### 4.3. Gait Pattern Overlap across Different Populations of Runners

Comparing gait patterns across the two populations of runners (Cohort 1 versus Cohort 2) showed that the gait patterns adopted by the young adult female runners in Cohort 2 were well-represented by both the in-lab and real-world data from Cohort 1 (males and females, age 18–59 years), suggesting that predictive models or inferences about gait developed from Cohort 1’s data would likely generalize well to Cohort 2. The depth-based distribution analysis used in this study could be used in future work to assess similar questions about generalizability of a predictive model or an experimental finding.

### 4.4. Comparison with the Previous Literature

When viewed in the context of previous work on in-lab versus overground or real-world running, this study’s results suggest that mean differences in gait metrics seen in other work (e.g., the kinematic changes seen in Lafferty et al. [6], the shifts in plantar pressure distribution seen in Hong et al. [36], or the decreased impact shocks seen during fatiguing overground running in García-Pérez et al. [37]) likely represent true distributional shifts, not merely changes in the average value of a gait parameter. Indeed, the success of machine learning-based classifiers that can accurately differentiate between running on pavement, synthetic track, and woodchip trails suggests that distributional shifts among gait metrics across different environments can be significant enough to produce very little overlap between running in different conditions [38]. Future work should investigate what environmental factors cause these distributional shifts in gait pattern.

### 4.5. Limitations

This study’s design had multiple limitations that should be considered alongside its primary findings regarding the degree to which in-lab gait patterns are well-representative of real-world gait patterns. First, though the inclusion criteria for Cohort 1 were designed to recruit a broad, diverse pool of runners in terms of age and running experience, it used convenience sampling, and thus cannot be considered a representative sample of all runners. Likewise, Cohort 2 represented a specific, homogenous group at risk for running-related injuries (female young adult runners); though gait data from Cohort 1 was well-representative of this specific population, the same finding may not be true for other specialized populations.

This study used a small set of gait metrics measured by consumer-grade wearable sensors to characterize a runner’s gait pattern. Though the metrics selected capture key parameters of running explained by simple models of gait (e.g., the mass-spring model [11]), these metrics still neglect aspects of gait that could be important for running gait assessment for runners at risk of injury, such as joint kinematics and kinetics [1].

Moreover, while the five gait metrics selected to represent a runner’s gait pattern in this study showed moderate to high correlations with gold-standard lab measurements (Appendix A), these device validations were carried out on in-lab running, not real-world running. As a consequence, true shifts in gait pattern between in-lab and real-world running might have been obscured by apparent differences caused by systematic errors in the wearable sensors which only occur during real-world running (e.g., if leg stiffness or ground contact time were measured with systematically greater error in the real world versus in the lab). These systematic differences could be caused by the different surfaces of in-lab vs. real-world running; not all real-world runs were completed on the same type of surface, and potential differences in surface stiffness between in-lab and real-world running could have affected the runner’s gait pattern. However, such differences would be expected to be magnified by environmental differences such as turns and inclines/declines; the fact that the findings of this study were substantively unchanged when stratifying real-world data to only include flat, straight segments provides some evidence that such systematic differences were not a major factor in the findings. 

A large portion of the in-lab vs. real-world differences were driven by vertical oscillation and leg stiffness, which are measured less accurately and, in the case of vertical oscillation, less consistently from device to device (see Appendix A, which details device-to-device differences in gait metric measurements). Though these device-based errors motivate future work using more precise equipment (e.g., video-based markerless motion capture), a sensitivity analysis did reveal that low in-lab/real-world overlap persisted even when iteratively removing leg stiffness and vertical oscillation as metrics in the gait pattern vector (see Figure 7). This robustness indicates that a substantial portion of the poor overlap in-lab/real-world is not attributable merely to inaccuracies in the measurement of any one individual gait metric.

This study used a treadmill to assess in-lab gait, but real-world running took place overground. Because of this design choice, it was not possible to disaggregate the effects of the treadmill itself from the effects of being in a gait laboratory versus in the real world. Future work should investigate in-lab treadmill, in-lab overground, and real-world overground running separately to parse out the effects of treadmill running from the in-lab environment itself. 

Advances in techniques for real-world assessment of gait could be used in future work to mitigate some of these limitations: using video-based gait assessment [39], research-grade wearable sensors [40], or fusing both of these measurement techniques together [41], could enable more comprehensive assessment of running gait patterns during both in-lab and real-world running, allowing for direct comparisons of a more complete representation of a runner’s gait pattern. Though these methods would not be scalable to the population sizes possible with consumer-grade wearable sensors, findings from smaller-scale studies with more intensive gait data could be used to confirm or refute the findings of the present study.

## 5. Conclusions

Research on the biomechanics of running is often limited to in-lab environments. If in-lab findings are to translate well to real-world applications in sport and clinical practice, the data collected in such in-lab studies needs to be representative of real-world, day-to-day training. This study’s findings indicate that a significant fraction of real-world running training involves gait patterns that are not well-represented by in-lab running. Over two-thirds of the steps a runner takes during real-world training are not well-represented by an in-lab data collection, even when this data collection is done throughout a prolonged run at a wide range of speeds, and even when real-world data are restricted only to flat, straight segments of runs.

One solution to the problems introduced by this distributional shift in gait patterns in the real world is to aggregate in-lab gait data from a large pool of runners so that predictive models or statistical inferences can “borrow” information across individuals. When such pools of data are used, real-world running gait patterns from a new runner are much better-represented by this aggregated pool of in-lab gait patterns—even when that new runner is drawn from a new subject population in a new geographic location.

The most immediate application of these findings in clinical practice is in the context of developing sensor-based, athlete-specific models of gait for monitoring training loads or implementing gait retraining protocols (e.g., the regression equations developed by Brund et al.) [42]. The findings of this study suggest that even an individualized model should be developed using a large pool of data across many runners to improve the proportion of real-world running that is well-represented by the in-lab data used to develop the model.

## Figures and Tables

**Figure 1 sensors-24-02892-f001:**
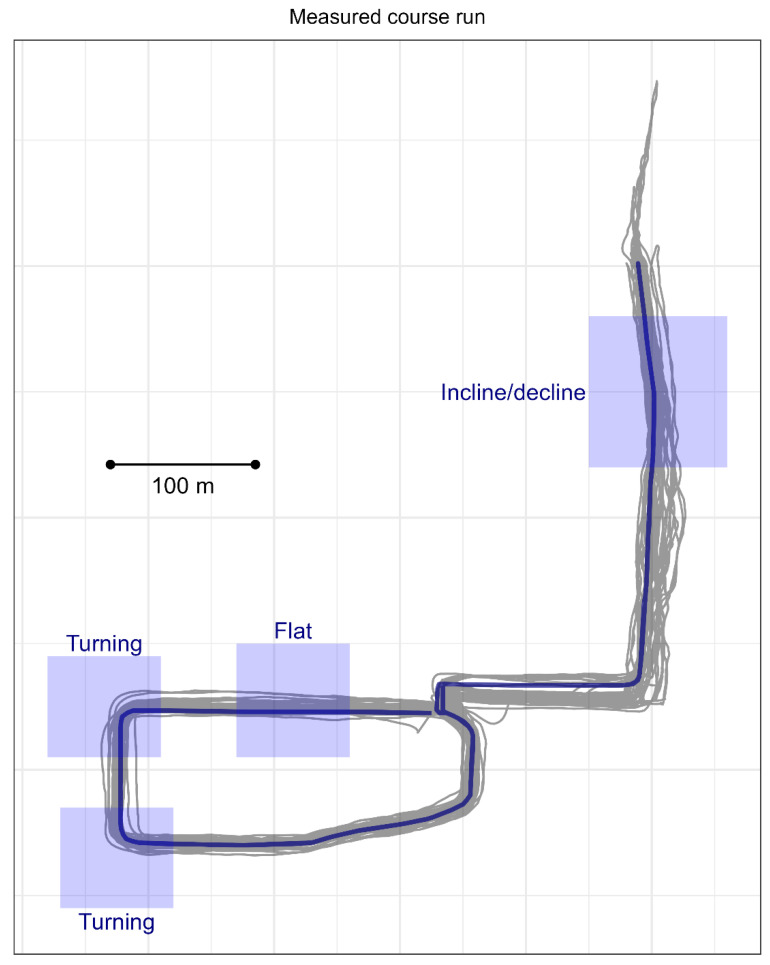
Route for 2.4 km measured course run completed by participants in Cohort 2. X- and Y-axes represent longitude and latitude. The route consisted of one counter-clockwise loop, an out-and-back segment on the incline/decline, and one clockwise loop. True route as determined using mapping software (OpenStreetMap) is shown in dark blue; actual global navigation satellite system (GNSS) data recorded by the participants are shown in gray. Each gray line represents course run completed by one subject. Shaded and labeled boxes indicate areas known to contain flat and straight running, left turns, right turns, inclines, and declines. Black line shows 100 m to scale.

**Figure 2 sensors-24-02892-f002:**
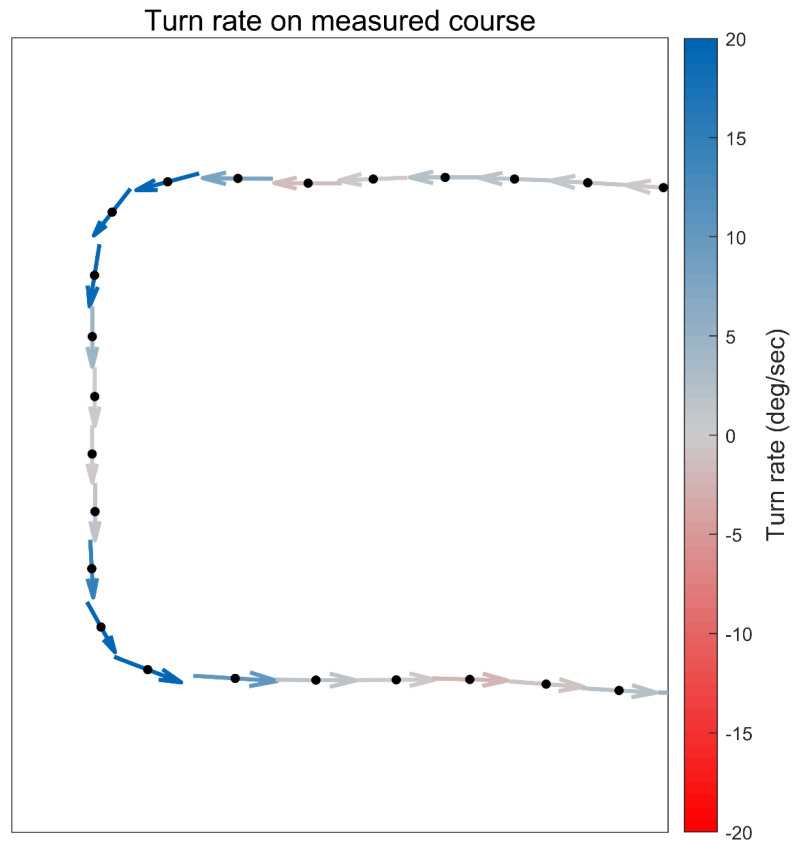
Illustration of turn rate data from one participant’s data during a portion of the known course run. At each global navigation satellite system (GNSS) location sample, the runner’s current heading is represented by a directional arrow. The central difference derivative of this heading is the runner’s current turn rate, in degrees per second; this turning rate is illustrated by the color of each heading arrow. In this case, the turn rate is positive, indicative of turning to the left (following the right-hand rule convention for vectors). For ease of visualization, the GNSS data have been downsampled by a factor of four in this figure.

**Figure 3 sensors-24-02892-f003:**
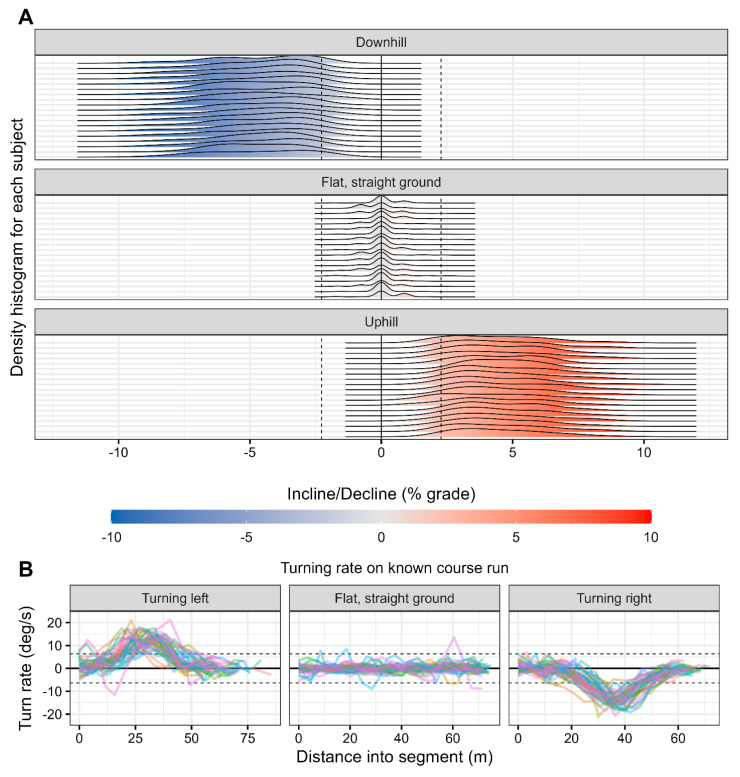
Illustration of applying threshold values to identify flat and straight running on segments of a known course run with turns, inclines, declines, and flat ground. (**A**) Subject-by-subject distribution of calculated incline/decline data from the known course run in Cohort 2 on known segments of downhill, flat and straight ground, and uphill running. Dashed lines show ±2.28% grade, the empirically-determined cut-off that retains >99% of running on the known flat, straight segment. (**B**) Subject-by-subject turn rates for the left turn, flat and straight, and right turn segments of the known course run in Cohort 2. Dashed lines show ±6.34 deg/s, the empirically-determined cut-off that retains >99% of running on flat, straight ground. For both the incline/decline and turn rate data, the known segments are clearly and reliably identified across subjects. Each line shows the trajectory of one participant from Cohort 2.

**Figure 4 sensors-24-02892-f004:**
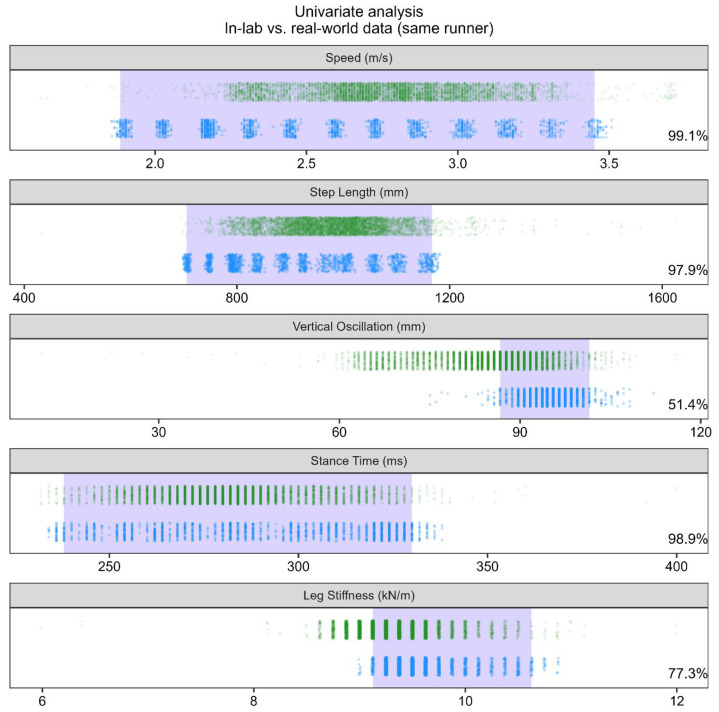
Example of univariate analysis applied to each of the five gait metrics from in-lab and real-world data from the same runner. The real-world data are represented by the green data points on the top portion of each plot panel, and the in-lab data are represented by the blue points on the bottom portion of each panel. Overlap is quantified as the proportion of real-world data which fall within the central 95% of the in-lab data (illustrated in blue shaded region).

**Figure 5 sensors-24-02892-f005:**
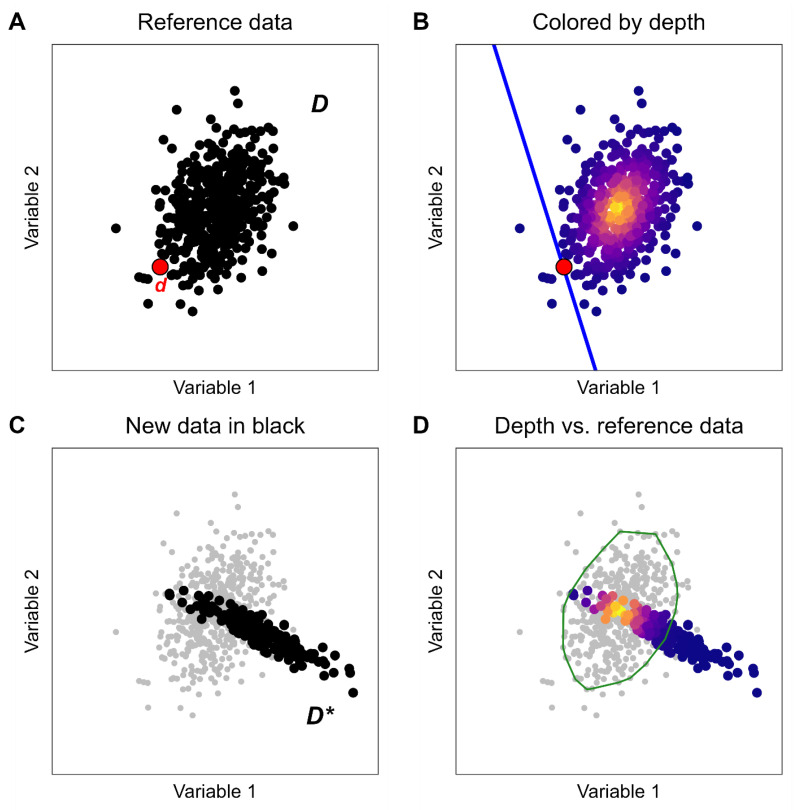
Illustrative example of half-space depth calculation for data in two dimensions (K=2). Given a reference distribution as a point cloud D (shown in panel (**A**)), the depth value of any point d (highlighted in red) relative to the reference distribution can be calculated as the minimum proportion of the reference data that can be “sliced off” by a half-space (in 2D, a line) that contains d (shown in panel (**B**)). In the example here, very few data points are sliced off by the half-space, and as such, the point d has a low depth value associated with it. In comparison, the points at the center of the point cloud (yellow) have higher depth values. The depth for each point in a new distribution $D^{*}$ can also be calculated with respect to the original reference distribution D (panel **C**). A point can be considered “well-represented” if it falls within the 95% convex hull of the reference distribution D. This convex hull is shown as the green shell in pane (**D**). The 95% convex hull contains the deepest 95% of the data from the reference distribution D. Though this figure illustrates depth using data in only two dimensions, the same methods can be applied to data in higher dimensions as well.

**Figure 6 sensors-24-02892-f006:**
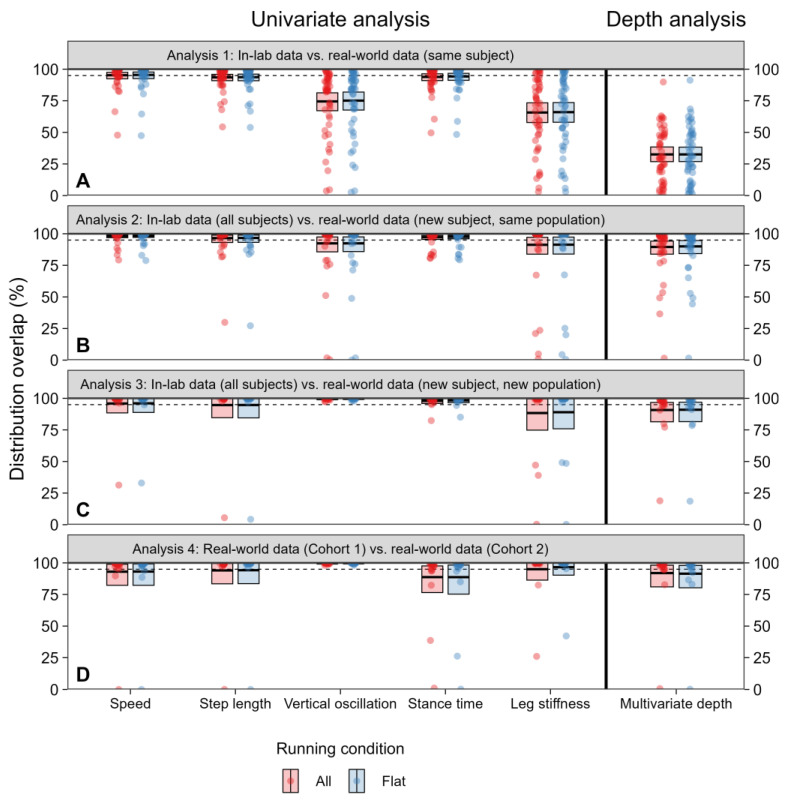
Distribution overlap results for each of the four analyses, using both the univariate analysis and the depth analysis. Each point represents data from one subject; black lines in crossbars represent the mean across subjects, and shaded regions in the crossbars represent bootstrapped 95% confidence intervals. Left panels show univariate analysis considering each gait metric separately; right panel shows depth analysis, which considers all gait metrics jointly. Dashed line shows the expected amount of overlap at the 95% confidence level (i.e., 95% overlap). Panel (**A**) shows overlap between one runner’s in-lab data (reference distribution) and that same runner’s real-world data (new distribution). Panel (**B**) shows overlap between in-lab data from all subjects and real-world data from a new runner from the same population (Cohort 1). Panel (**C**) shows overlap between in-lab data from all subjects and real-world data from a new runner from a new population (Cohort 2). Panel (**D**) shows overlap between real-world data from one population (Cohort 1) and real-world data from another population (Cohort 2). For all analyses, stratifying real-world data to only include running on flat, straight segments resulted in <5% change in distributional overlap (blue vs. red points and shaded crossbars).

**Figure 7 sensors-24-02892-f007:**
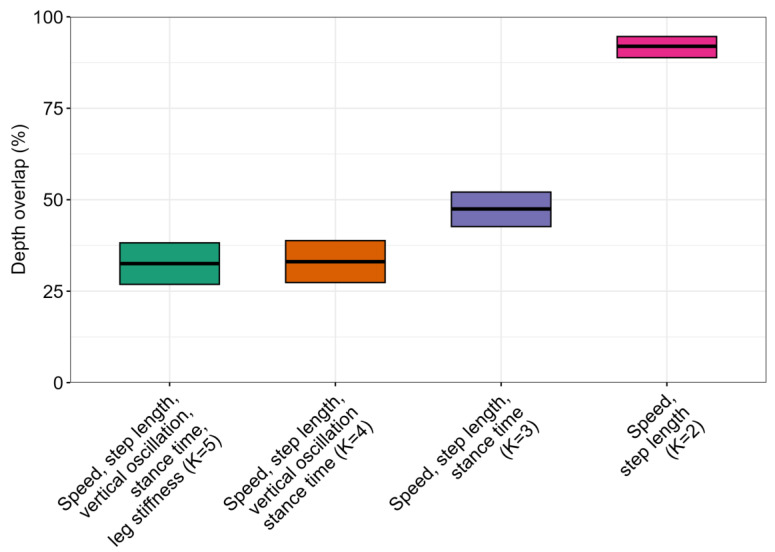
Sensitivity analysis of in-lab versus real-world data. Analysis shows effects of iteratively reducing the number of gait metrics used to represent the runner’s gait pattern. Black bar shows mean overlap across subjects, and shaded box shows bootstrapped 95% confidence interval for the mean. The left-most column (K=5) is the original analysis (Analysis 1). This analysis shows that leg stiffness and vertical oscillation drive some of the poor overlap between in-lab and real-world running, but even when gait pattern is characterized only by speed, step length, and stance time, mean overlap remains below 50%. The high overlap when considering only speed and step length (K=2) indicates that a change in speed–step length strategy (i.e., how runners modulate their cadence and step length to achieve a given speed) cannot explain the poor overlap between in-lab and real-world running.

**Figure 8 sensors-24-02892-f008:**
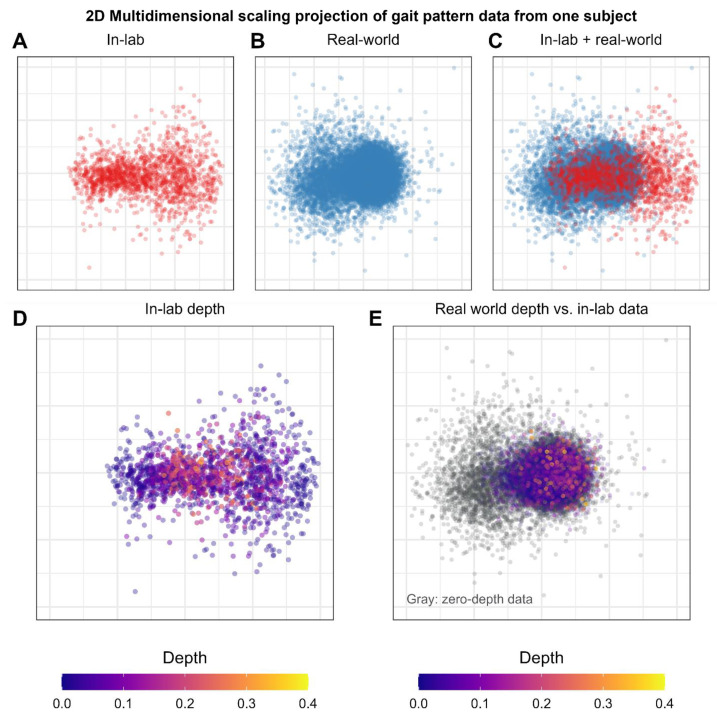
Visualization of the five-dimensional gait pattern for one subject in Cohort 1, projected to two dimensions using multidimensional scaling, a dimensionality reduction technique [33]. Panels (**A**–**C**) show the in-lab (red) and real-world (blue) gait pattern data for this runner. Some distributional overlap exists, but this runner’s real-world gait pattern shows a clear distributional shift away from the in-lab distribution. When using the in-lab data as a reference distribution for depth analysis (panel (**D**)), only 30.1% of this runner’s real-world data (panel (**E**)) fall within the 95% depth threshold of the in-lab data distribution. For this runner, over half of the real-world data have a depth of zero (shown as gray data points in panel (**E**), meaning they are completely outside of the distribution of gait patterns seen during in-lab running.

**Table 1 sensors-24-02892-t001:** Reference distribution and new data distribution for each analysis.

Analysis	Comparison	Reference Distribution	New Data Distribution
Analysis 1	In-lab vs. real-world running (same runner)	Cohort 1 in-lab data from one subject	Cohort 1 real-world data from same subject
Analysis 2	In-lab vs. real-world running (new runner, same population)	Cohort 1 in-lab data from all but one subject	Cohort 1 real-world data from one left-out subject
Analysis 3	In-lab vs. real-world running (new runner, new population and location)	Cohort 1 in-lab data from all subjects	Cohort 2 real-world data from all subjects
Analysis 4	Real-world running in one population vs. real-world running in new population and location	Cohort 1 real-world data from all subjects	Cohort 2 real-world data from all subjects

**Table 2 sensors-24-02892-t002:** Participant demographics from both cohorts.

Cohort 1 (N = 49 Participants).					
	Min	1st Quartile	Median	3rd Quartile	Max
Age (year)	18	24	29	35	58
Height (m)	1.54	1.68	1.74	1.81	1.93
Mass (kg)	43	60.9	66.7	79.2	110.5
Body Mass Index (kg/m^2^)	17.49	20.51	22.3	24.35	31.6
Training volume (km/wk)	16.09	24.14	40.23	51.5	88.51
Experience (year)	2	7	10	15	47
Sex	25 M, 24 F				
Category (self-reported)	Novice: 0, Recreational: 29, Competitive: 18, Elite: 2				
**Cohort 2 (N = 19 Participants).**					
	**Min**	**1st Quartile**	**Median**	**3rd Quartile**	**Max**
Age (year)	18	19	20	21	32
Height (m)	1.52	1.59	1.66	1.69	1.77
Mass (kg)	47	53.35	57.3	63.7	80.8
Body Mass Index (kg/m^2^)	18.35	20.74	22.05	22.64	28.09
Training volume (km/wk)	12.87	21.73	32.19	41.04	112.65
Experience (year)	2	6	7	9	18
Sex	0 M, 19 F				
Category (self-reported)	Novice: 0, Recreational: 14, Competitive: 4, Elite: 1				

## Data Availability

Code and anonymized data supporting the findings of this study are available on GitHub (https://github.com/johnjdavisiv/inlab-vs-realworld) and FigShare (https://doi.org/10.6084/m9.figshare.23662662).

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
