# Peer review of "Are Gait Patterns during In-Lab Running Representative of Gait Patterns during Real-World Training? An Experimental Study"

_sensors, 2024, doi:10.3390/s24092892_

Round 1
Reviewer 1 Report
Comments and Suggestions for Authors
The author(s) has presented an observational study on “gait patterns during in-lab running representative of gait patterns during real-world training”. The study includes two cohorts equipped with consumer-grade wearable sensors measuring speed, step length, vertical oscillation, stance time, and leg stiffness. The paper is well-written and organized well. However, the methodology and results are partially justified, and a full revision of the paper is required. The author(s) should consider the following review points before submitting a revision.
1. The paper does not provide a clear and accurate description of the problem in the introduction section.
2. The deliverables of the research should be mentioned in the Introduction section and also highlight the robot-based assistant.
3. Provide a tabular or graphical description of a problem addressed.
4. The authors are suggested to highlight how the work is different from existing literature since it is not very evident.
5. Provide the specification of the sensor used, and data description including no of the subject, age group etc. How many IMU sensors were considered and their position?
6. The deliverables of the research should be mentioned in the Introduction section and also highlight the robot-based assistant.
7. How the angle is being calculated from IMU data linear acceleration provides an Inverse kinematic solution.
8. Also, please provide the raw trajectories of hip and knee and ankle joints.
9. Which tool is used for the validation of trajectories?
10. Please specify the number of subjects, sensor placement, and calibration of the sensor.
11. Please compare the work with the previous state of artwork.
12. The limitations of the research work should be mentioned in the Conclusion.
13. Many of the references are over last 5 years old. It is expected that most of the references should be within last 3-5 years and provide a clear motivation of work. Please update the same. Refer these works:
(i) Semwal, Vijay Bhaskar, et al. "Speed, cloth and pose invariant gait recognition-based person identification." Machine learning: theoretical foundations and practical applications. Singapore: Springer Singapore, 2021. 39-56.
(ii) Semwal, Vijay Bhaskar, et al. "Development of the LSTM Model and Universal Polynomial Equation for all the Sub-phases of Human Gait." IEEE Sensors Journal (2023).
(iii) Gaud, Neha, Maya Rathore, and Ugrasen Suman. "Human gait analysis and activity recognition: A review." 2023 IEEE Guwahati Subsection Conference (GCON). IEEE, 2023.
Reviewer 2 Report
Comments and Suggestions for Authors
The paper, titled "Are Gait Patterns During In-Lab Running Representative of Gait Patterns During Real-World Training? An Observational Study," is engaging and delves into the investigation of distributional differences in gait patterns between in-lab and real-world running using consumer-grade wearable sensors. The primary objective is to ascertain the accuracy of in-lab data in representing gait during free-living training.
While some gait metrics, including speed, step length, and stance time, exhibit consistency when transitioning from lab to real-world settings, the paper highlights a significant shift in the overall distribution of gait patterns. As important, the findings reveal that fewer than one-third of steps taken during real-world running closely align with an individual's in-lab data, even when the analysis is limited to flat, straight segments.
I appreciate the paper overall, and my only suggestion would be to improve the introduction by providing a better context regarding prior research on this topic. Offering insights into existing studies and explaining why the authors chose to explore this particular analysis would provide readers with a better understanding of the paper's significance and research gap. Completing the introduction, I think the authors will find a few more items that would be worth quoting.
Referring to the tables, I suggest the authors remove the appendices and supplementary material and put everything in the text. Such a step will make it easier to read.
Going through the specific questions required by the magazine:
· Is the manuscript clear, relevant for the field and presented in a well-structured manner?
Yes, the manuscript is clear and well-structured. I recommend incorporating the content from the appendix and supplementary material into the main text, as I mentioned earlier.
· Is the manuscript scientifically sound and is the experimental design appropriate to test the hypothesis? The introduction needs completion in this regard.
Reviewer 3 Report
Comments and Suggestions for Authors
Thank you for the opportunity to review this interesting manuscript. While I am confident the readership of Sensors would find some of the content interesting, the manuscript is difficult to follow due to the sheer amount of data presented. It is also biased towards literature which suggests there are no differences in gait when observed in a laboratory condition compared to real-life. There is plenty of research to refute this, which the authors have not presented nor considered.
Please find my detailed comments below:
Abstract
It is unclear whether all the distances run under all conditions were the same i.e. 2.4km
Please consider expanding the final sentence to highlight the benefit of being able to predict real-world gait behaviour.
Introduction
The introduction describes the potential benefits of this research in clinical and research terms, but has not considered the role it may play in professional sport. Please consider adding this information.
The introduction is biased towards research that has failed to identify differences between in-lab and real-life gait parameters, even though there is a wealth of information in the literature on the biomechanical differences to gait in different conditions. Please consider this literature too, and clearly describe the justification for this research with reference to the current literature.
Methodology
It is unclear why the two cohorts underwent different protocols. Please also provide further information on the ‘real-world’ runs. Were these a specific distance/terrain? Please also justify why the second group was female only.
Please describe how comparable the data from the foot pod and GNSS are, given that they were used interchangeably.
There is no mention on whether ethical approval was granted for this study in the methodology.
How was the runner’s preferred running speed determined? Were they required to know this information prior to taking part in the study?
The statistical section needs to be significantly improved, as it currently reads like an introduction or discussion. The methods should simply present what was done. It is also unusual to see some results within the methodology. Please consider removing these.
Please describe how the sample sizes were determined.
Results
Why were the two groups so different in size? How might this impact the statistical analyses performed on the data?
The sheer volume of the data presented in this section makes the entire manuscript difficult to follow. It is also unclear what the main objectives of the research are when so much information is being presented. Please consider presenting data that is relevant to the research question only.
Discussion
The discussion does not adequately describe the study results with regards to previous studies. It is therefore difficult to interpret the data presented. Please ensure that all presented results are discussed appropriately.
Conclusion
The authors state ‘Research on the biomechanics of running often operates under the assumption that a runner’s gait during an in-lab motion capture analysis is representative of real-world, day-to-day training’. However, I am not convinced that this is true. Biomechanists generally appreciate the differences between lab and real-life conditions, but none of this literature is not presented in the manuscript. Please reword the conclusion accordingly and ensure that you are presenting your data in an unbiased manner.
General comment
This is a very long manuscript. Please could the authors check the guidelines of this journal, as they may be required to significantly reduce the wordcount and number of figures prior to publication. In particular, much of the methodology could be much more succinct, as there are discussions on the pros and cons of various statistical analyses within the methods. While these are important to consider, they are much better placed in the Discussion. The results are also difficult to follow and interpret given the volume of data presented.
Round 2
Reviewer 1 Report
Comments and Suggestions for Authors
Though the paper is partially revised. The author(s) has ignored many important reviewer comments. it is suggested not to ignore a single comment.
The literature review table is not included
still, results are partially justified and relevant work of gait in literature review is not considered:
Semwal, Vijay Bhaskar, et al. "Speed, cloth and pose invariant gait recognition-based person identification." Machine learning: theoretical foundations and practical applications. Singapore: Springer Singapore, 2021. 39-56. (ii) Semwal, Vijay Bhaskar, et al. "Development of the LSTM Model and Universal Polynomial Equation for all the Sub-phases of Human Gait." IEEE Sensors Journal (2023). (iii) Gaud, Neha, Maya Rathore, and Ugrasen Suman. "Human gait analysis and activity recognition: A review." 2023 IEEE Guwahati Subsection Conference (GCON). IEEE, 2023.
